# Longitudinal Consumption of Ergothioneine Reduces Oxidative Stress and Amyloid Plaques and Restores Glucose Metabolism in the 5XFAD Mouse Model of Alzheimer’s Disease

**DOI:** 10.3390/ph15060742

**Published:** 2022-06-13

**Authors:** Clayton A. Whitmore, Justin R. Haynes, William J. Behof, Adam J. Rosenberg, Mohammed N. Tantawy, Brian C. Hachey, Brian E. Wadzinski, Benjamin W. Spiller, Todd E. Peterson, Krista C. Paffenroth, Fiona E. Harrison, Robert B. Beelman, Printha Wijesinghe, Joanne A. Matsubara, Wellington Pham

**Affiliations:** 1Vanderbilt University Institute of Imaging Science, Vanderbilt University Medical Center, Nashville, TN 37232, USA; clayton.whitmore@vumc.org (C.A.W.); j.r.haynes@vumc.org (J.R.H.); william.j.behof@vumc.org (W.J.B.); adam.j.rosenberg@vumc.org (A.J.R.); n.tantawy@vumc.org (M.N.T.); todd.e.peterson@vumc.org (T.E.P.); 2Department of Radiology and Radiological Sciences, Vanderbilt University Medical Center, Nashville, TN 37232, USA; 3Department of Biochemistry, Vanderbilt University, Nashville, TN 37232, USA; brian.c.hachey@vanderbilt.edu; 4Department of Pharmacology, Vanderbilt University, Nashville, TN 37233, USA; brian.wadzinski@vanderbilt.edu (B.E.W.); benjamin.spiller@vanderbilt.edu (B.W.S.); krista.c.paffenroth@vanderbilt.edu (K.C.P.); 5Vanderbilt Brain Institute, Vanderbilt University, Nashville, TN 37232, USA; fiona.harrison@vumc.org; 6Department of Medicine, Diabetes, Endocrinology & Metabolism, Vanderbilt University Medical Center, Nashville, TN 37232, USA; 7Vanderbilt Memory and Alzheimer’s Center, Vanderbilt University Medical Center, Nashville, TN 37212, USA; 8Department of Food Science, Center for Plant and Mushroom Foods for Health, Penn State University, University Park, PA 16802, USA; rbb6@psu.edu; 9Department of Ophthalmology and Visual Sciences, University of British Columbia, Vancouver, BC V5Z 3N9, Canada; printha.wijesinghe@ubc.ca (P.W.); joanne.matsubara@ubc.ca (J.A.M.); 10Department of Biomedical Engineering, Vanderbilt University, Nashville, TN 37235, USA; 11Vanderbilt Ingram Cancer Center, Nashville, TN 37232, USA; 12Vanderbilt Institute of Chemical Biology, Vanderbilt University, Nashville, TN 37232, USA; 13Vanderbilt Institute of Nanoscale Science and Engineering, Vanderbilt University, Nashville, TN 37235, USA

**Keywords:** ergothioneine, ROS, PET, Alzheimer, oxidative stress, 5XFAD

## Abstract

*Background:* Ergothioneine (ERGO) is a unique antioxidant and a rare amino acid available in fungi and various bacteria but not in higher plants or animals. Substantial research data indicate that ERGO is a physiological antioxidant cytoprotectant. Different from other antioxidants that need to breach the blood–brain barrier to enter the brain parenchyma, a specialized transporter called OCTN1 has been identified for transporting ERGO to the brain. *Purpose*: To assess whether consumption of ERGO can prevent the progress of Alzheimer’s disease (AD) on young (4-month-old) 5XFAD mice. *Methods and materials*: Three cohorts of mice were tested in this study, including ERGO-treated 5XFAD, non-treated 5XFAD, and WT mice. After the therapy, the animals went through various behavioral experiments to assess cognition. Then, mice were scanned with PET imaging to evaluate the biomarkers associated with AD using [^11^C]PIB, [^11^C]ERGO, and [^18^F]FDG radioligands. At the end of imaging, the animals went through cardiac perfusion, and the brains were isolated for immunohistology. *Results*: Young (4-month-old) 5XFAD mice did not show a cognitive deficit, and thus, we observed modest improvement in the treated counterparts. In contrast, the response to therapy was clearly detected at the molecular level. Treating 5XFAD mice with ERGO resulted in reduced amyloid plaques, oxidative stress, and rescued glucose metabolism. *Conclusions*: Consumption of high amounts of ERGO benefits the brain. ERGO has the potential to prevent AD. This work also demonstrates the power of imaging technology to assess response during therapy.

## 1. Introduction

Alzheimer’s disease (AD) is the most common cause of dementia among elderly people, but the precise factors contributing to its etiology have not yet been determined [1]. In an aging population, dementia will present one of the greatest challenges to society in this century. The mechanisms that regulate neuronal degeneration in AD remain unclear. However, the cytopathological hallmarks of AD appear to be the deposition of extracellular amyloid-β (Aβ) plaques between neurons and the intracellular accumulation of phosphorylated (p)-Tau, which ultimately lead to profound neuronal toxicity and atrophy [2]. As Aβ plaque formation is one of the underlying mechanisms implicated in AD, prevention of such formation, particularly at early disease onset, would be an important goal to eradicate this disease.

Despite decades of research, the pathogenetic mechanisms that initiate Aβ aggregation are still largely unknown. Postmortem studies of AD human brains showed that there is a huge imbalance of oxidant–antioxidant status that leads to cell death and tissue damage [3]. Reactive oxygen species (ROS) play a pivotal role in normal cellular and signaling pathways, albeit excessive generation of ROS leads to harmful effects, including cellular and protein damage [4,5,6]. Oxidative stress and damage due to ROS have been implicated in the pathogenesis of AD [7]. Extensive studies have highlighted in particular the role of superoxide anion (^•^O_2_^−^), hydroxyl radical (^•^OH), hydrogen peroxide (H_2_O_2_), and nitric oxide (NO^•^) in the oxidative-stress-mediated neurodegeneration of AD [8,9]. The brain is more vulnerable to oxidative stress than any other organ, since the components of neurons, such as lipids, proteins, and nucleic acids, can be oxidized in AD. These changes in oxidative stress are associated with increased metal levels, inflammation, and Aβ peptides [10]. Postmortem analysis of human AD brains showed that iron (Fe), zinc (Zn), and copper (Cu) are associated with oxidative stress, and they accumulate in the Aβ plaques of these AD patients [11,12]. Others have also demonstrated that these metals have a very high affinity for Aβ-42, a major component of amyloid plaques [13]. Notably, these three metals are cofactors in redox reactions, and they can switch between oxidation states through the Fenton mechanism to generate radicals, which catalyze the conversion of inert peptide side chains into very reactive species. For example, Cu^+^, Fe^2+^, and Zn^+^ could (i) act as cofactors with enzymes to convert inert peptide side chains to active and unstable species; (ii) react with H_2_O_2_ under stress conditions to generate ^•^OH, which in turn induces particular aggregation, ultimately leading to the formation of annular protofibrils and fibrillar Aβ aggregation [14]. This formation of structural aggregates with biophysical properties is associated with high neurotoxicity and neurodegeneration. Control or suppression of these reactions may, therefore, provide a valid method to slow down disease onset or Aβ accumulation and associated damage.

Mushrooms are a rich source of four bioactive compounds essential to human health, including selenium [15,16], vitamin D [17,18], glutathione [19], and ergothioneine (ERGO) [20,21]. Except for Vitamin D, the remaining compounds are antioxidant agents. Accumulating evidence indicates that ERGO in particular is a physiological antioxidant cytoprotectant [22,23,24,25], existing in the body as a water-soluble zwitterion (Figure 1A). ERGO protection against cytotoxicity elicited by Cu^2+^, H_2_O_2_, Fe, and sodium nitrite (NaNO_2_) [26,27,28,29] is derived from its conspicuous affinity for metal cations, such as Fe and Cu, permitting the capture and neutralization of associated radicals [30]. Humans obtain ERGO through the diet, but blood ERGO concentration decreases significantly with age, and markedly lower levels have been found in individuals with mild cognitive impairment compared to healthy counterparts. This observation supports the concept that ERGO deficiency acts as a risk factor for neurodegeneration [31].

To test the biodistribution and pharmacokinetics of ERGO in vivo, we recently reported the development of an [^11^C]ERGO PET radioligand [32]. Our study suggested that this zwitterion molecule is very easy to formulate for oral administration, since it is very polar. Based on our in vivo PET imaging data, we hypothesized that ERGO could serve as an ideal antioxidant for AD, since it can be distributed to the brain by an oral route. ERGO uptake is very high in the small intestine, suggesting that there are abundant receptors (OCTN1 (novel organic cation transporter 1)) in the gut that would shuttle the compound to the circulation where it will be distributed to the brain mediated by OCTN1 receptors expressed in the brain parenchyma.

Since ROS and metal products have a profound impact on downstream Aβ pathology, in this work, we investigated whether consumption of ERGO benefits the brain of 5XFAD mice. Our working hypothesis is that this dual ROS/metal scavenging antioxidant will serve as a potent radical inhibitor to prevent Aβ aggregation (Figure 1B). The data showed a promising effect of longitudinal intake of ERGO when testing the compound on 5XFAD mice. The treated cohort of 5XFAD mice showed modest improvement in cognition. In contrast, we observed a marked reduction in biomarkers related to AD pathogenesis at the molecular level using PET imaging. In particular, Aβ burden was substantially reduced, and glucose metabolism was rescued in treated young 5XFAD mice. These observations have been reported in the past with the Caenorhabditis elegans model [33] or C56BL/6J mice [34], albeit this is the first time we demonstrated in vivo imaging to track the response of these biomarkers to therapy on a 5XFAD model. Further, this work also suggests that the [^11^C]ERGO PET radioligand could potentially be used to track oxidative stress levels as a biomarker to assess the efficacy of AD therapy.

## 2. Results

### 2.1. ERGO Is a Metal Scavenger

To demonstrate that ERGO has a high affinity for metals, we developed an assay utilizing 2D-HPLC equipped with a nickel column. In principle, nickel is a transition metal like Fe, Cu, and Zn. The nickel column can immobilize hexahistidine-tag (His-tag) recombinant proteins, such as recombinant antibodies harboring a His tag (His-Abs). In our experimental design, nickel forms a substantial coordination equilibrium with histidines via the imidazole ring structures to form a stable metal/ligand complex. This assay uses two different buffer reservoirs, a wash buffer (PBS) and an elution buffer (PBS containing 50 mM imidazole). To remove the His-Abs (14 KDa) from the nickel column, the washing buffer was switched to the elution buffer, and the His-Abs dislodged from the column were detected by the 2D-HPLC (Figure 1C, left). In a different phase of the experiment, the His-Abs solution was spiked with ERGO (14.5 mM) prior to injection into the HPLC system. As a result, the His-Abs were eluted from the nickel column immediately by the washing buffer (Figure 1C, right). Overall, the data suggest that with a strong affinity with metals, ERGO competes with the His-tag and purges the His-Abs from the column. The integrity of the eluted His-Abs was collected and confirmed by orbit trap mass spectrometry (Appendix A).

We also assessed ERGO’s transition metal scavenging using cupric ion chelating (CIC) and ferrous ion chelating (FIC) assays. In these experiments, pyrocatechol violet or ferrozine forms a complex with free Cu(II) or Fe(II), respectively, resulting in a chromophore with a strong absorbance signal. The pyrocatechol violet-Cu(II) complex has an absorbance lambda max (λ_max_) at 630 nm, and ferrozine-Fe(II) has a λ_max_ at 562 nm. In the presence of ERGO, fewer free metals are available to form these complexes, resulting in less absorbance. Data in Figure 1D,E show that the signals decreased as the concentrations of ERGO increased in a dose-dependent fashion.

### 2.2. ERGO Is a ROS Scavenger

We used a commercially available DCFH-DA (2′,7′-dichlorodihydrofluorescein diacetate) probe to assess the ROS scavenging power of ERGO in HeLa and neuroblastoma SH-SY5Y cells. Upon penetrating the cytoplasm, DCFH-DA is deacetylated by endogenous esterases to a non-fluorescent intermediate, followed by oxidation by free radicals. This generates a continuous and extended conjugation system, resulting in the production of highly fluorescent DCF (2′,7′-dichlorofluorescein). In an aqueous condition, DCF has a maximal excitation and emission of 495 nm and 529 nm, respectively. Well-adhered HeLa or SH-SY5Y cells previously incubated with DCFH-DA were exposed to a commercial ROS initializer and ERGO concentrations ranging from 0 to 100 mM. The data in Figure 1F,G suggest that ERGO scavenges ROS products, leading to the quenching of the fluorescence signal of the DCF probe. The scavenging effect is dose dependent; starting from 5 mM, ERGO attenuates the fluorescence significantly. At a dose of 100 mM, the fluorescence is nearly negligible. Each point in the assay was measured in pentaplicate, *p* < 0.0001.

### 2.3. Timeline of the Therapy and Processing

To address our primary question of whether longitudinal consumption of ERGO as a potent antioxidant benefits the brain, three cohorts of animals (*n* = 12 non-treated WT control, *n* = 12 non-treated 5XFAD, and *n* = 18 ERGO-treated 5XFAD) were used (Figure 2A). Starting at the age of 8 weeks, the animals were treated with high doses of ERGO solution (25–50 mg/Kg) formulated in double-distilled (dd) water via oral gavage three times a week over the course of 8 weeks. To reduce bias in the behavioral experiments with regard to the potential stress during gavage treatment, non-treated animals underwent an identical gavage process with the vehicle, water only. The efficacy of the therapy was evaluated by behavioral testing, followed by three different non-invasive PET imaging sessions to assess the biomarkers related to AD pathology. In addition to assessing Aβ levels and glucose metabolism using the [^11^C]PIB and [^18^F]FDG probes, respectively, we also tested for the first time whether oxidative stress could be used as an indicator of response to therapy using the [^11^C]ERGO PET radioligand. Finally, the animals were euthanized, and the brains were collected for immunohistochemical analysis.

### 2.4. ERGO Treatment Prevents Early Cognitive Deficits in 5XFAD Mice

At the end of the ERGO treatment, the mice were transferred to the animal neurobehavior facility to acclimate for at least three days before behavioral assessment. The cognition of WT (*n* = 12), non-treated 5XFAD (*n* = 12), and ERGO-treated 5XFAD (*n* = 18) mice was assessed using a variety of different assays.

We first confirmed that ERGO treatment had no adverse effects on activity or anxiety-like behavior in the animals that could potentially impact the efficacy and translatability of ERGO as an interventional strategy or that would impact the interpretation of cognitive behavioral tasks. We used the elevated zero maze (EZM) to assess anxiety-like behaviors of 5XFAD mice following ERGO treatment. All mice had equivalent exploration in the maze, as assessed by total distance traveled (F_(2, 39)_ = 0.82, *p* = 0.45). Anxiety-like behavior in this task is indexed via exploration of brighter, open areas of the maze compared to darker, enclosed regions, perceived as “safe zones” [35]. Each group spent close to 60% of their time exploring the closed zones, suggesting normal exploratory activity (F_(2, 39)_ = 0.19, *p* = 0.82) (Figure 2B). Exploratory activity in a novel environment was further assessed in locomotor activity chambers across a 30 min session. Although 5XFAD mice are sometimes reported to be hyperactive at 8 months and older, we observed similar exploration in all three groups (F_(2, 39)_ = 0.80, *p* = 0.46) (Figure 2C). All mice were able to learn the rotarod, as evidenced by increasing latencies to fall across the three days of testing (F_(2, 63__)_ = 24.13, *p* < 0.0001), suggesting intact cerebellum-dependent motor learning. There were no differences among the groups or day x group interaction (F_s_ < 1.20, *p* > 0.32) (Figure 2D). Exploration in the empty novel object arena was equivalent among groups (F_(2, 37)_ = 0.24, *p* = 0.78, data unavailable for two mice due to computer tracking error). Mice that explored objects for less than 10 s in total or explored any single object for less than 5 s were not included in recall analyses (three non-treated 5XFAD, two treated 5XFAD). For all other mice, the initial exploration did not vary according to location (left/right) or group (F_s_ < 1.80, *p* > 0.18) and ranged from 20.1 to 77.8 s in WT, 14.7 to 124.9 s in non-treated 5XFAD, and 11.5 to 124.6 s in ERGO-treated 5XFAD. During the final test, trial data from one additional 5XFAD mouse were excluded because it spent 0 s within the target zone. However, most mice spent more time in proximity to the novel object than the familiar object, but the discrimination index did not vary among the groups (F_(2, 32)_ = 0.48, *p* = 0.62) (Figure 2E).

Mice were then trained to associate a series of three small (0.5 mV, 1 s) electric shocks with the termination of a 30 s auditory cue in the conditioned fear task. Recall of the shocks was tested by indexing time spent freezing (a fear response), when mice were exposed to the same testing context, or to the previously paired tone within a novel testing context 24 and 25 h following training, respectively. All mice showed a strong memory of the testing environment, as evidenced by similar levels of freezing when returned to the original testing chambers (Kruskal–Wallis statistic = 0.54, *p* = 0.77) (Figure 2F). When mice were exposed to a novel context (plastic-lined walls and floor and vanilla odor), the exploratory activity resumed, indicating that prior freezing, and therefore learning, was specific to the testing context. In contrast, when the cue was re-introduced, WT control and ERGO-treated 5XFAD mice showed improved recall compared to non-treated 5XFAD, as evidenced by increased freezing during presentation of the cue (Figure 2G). Although all mice increased freezing time when the cue was played compared to the “tone-off” portion of the trial (F_(1, 37)_ = 65.95, *p* < 0.0001), performance was not equivalent between the groups. ERGO-treated 5XFAD mice strongly resembled the performance of control WT mice, with freezing close to 50% of the time that the cue was played (*p* < 0.0001) compared to non-treated 5XFAD control animals that froze only approximately 25% of the time (*p* < 0.05).

### 2.5. Longitudinal Consumption of ERGO Mitigates Aβ Aggregation in Young 5XFAD Mice

Non-invasive, in vivo, and robust assessment of Aβ levels in the brains of treated and non-treated 5XFAD mice were performed using the [^11^C]PIB PET probe, the chemical structure of which has been described elsewhere [36]. All animals were injected via the tail vein (Appendix A) with the same radioisotope dose (~15 MBq of [^11^C]PIB) and volume (100 μL/mouse). To test the preventive effect of ERGO, we treated 5XFAD mice starting at the age of 2 months. The idea of this work is based on the premise that young 5XFAD mice do not develop Aβ deposits/plaques in the brains until 2 month old [37]. We found a steady increase in PET signal in non-treated 5XFAD mice (*n* = 5), at the age of 4 months (Figure 3D–F), suggestive of the presence of extracellular Aβ plaques. The PET signals obtained from ERGO-treated, age-matched 5XFAD (*n* = 7) (Figure 3G–I) were very weak and resembled the background signal found in WT mice (*n* = 4) (Figure 3A–C). We co-registered the PET/CT image to an MRI template for quantitative PET analysis of regional uptake in the brain. The volumetric region of interest was drawn around the cortex, hippocampus, striatum, thalamus, and cerebellum, in addition to the whole brain. The concentration of the injected [^11^C]PIB in PET imaging expressed as percentage of injected dose per gram tissue (%ID/g) was significantly higher in every subregion of the brain (*p* < 0.05) in non-treated 5XFAD compared to treated counterparts (~30–50%) (Figure 3J).

To confirm the in vivo observations, animals were perfused, and the brain sections were stained with anti-Aβ antibody, 6E10. Our staining data showed that in non-treated 5XFAD mice, Aβ was highly expressed in the brain regions typically susceptible to AD, including the hippocampus, cortex, and the thalamus (Figure 3K,N,Q), at approximately 4 months of age. In particular, we observed a clear difference in the hippocampus, pyramidal cortex, and thalamus of non-treated compared to ERGO-treated 5XFAD mice (Figure 3L,O,R). The pixel counts were further quantified between cohorts with ImageJ; this demonstrated a 4-fold (*p* < 0.05), 5-fold (*p* = 0.0004), and 2-fold (*p* < 0.05) decrease in Aβ level in the regions of the hippocampus, pyramidal cortex, and thalamus, respectively, in treated 5XFAD mice (Figure 3M,P,S). (Appendix A).

### 2.6. [^11^C]ERGO PET Radioligand Detects Oxidative Stress Reduction in ERGO-Treated 5XFAD Mice

Recently, our [^11^C]ERGO PET radioligand has been demonstrated as an imaging biomarker for oxidative stress in a mouse model of lipopolysaccharides (LPS) [32] and 5XFAD mice [38]. Thus, in this study, [^11^C]ERGO PET radioligand was used to assess the fluctuation in oxidative stress with respect to the ERGO therapy. We observed a very modest reduction in PET signal in a 20 min dynamic scan after intravenous (IV) injection, pertaining to oxidative stress in the brains of the treated mice (Figure 4D–F) compared to non-treated 5XFAD mice (Figure 4A–C). This observation could be due to the fact that the animals were treated with high doses of ERGO during gavaging. Although we stopped treating mice with ERGO weeks before injection of the [^11^C]ERGO PET radioligand, the residual ERGO from the last treatment could have blocked the majority of the binding sites and thus resulted in less-to-none PET signal. Nevertheless, post-imaging data analysis revealed that [^11^C]ERGO uptake differentiated the level of oxidative stress of non-treated 5XFAD mice from the age-matched treated 5XFAD mice (Figure 4G) (*p* < 0.05).

We also found that other key inflammation-related protein markers were positively corelated with [^11^C]ERGO PET imaging data, supporting the [^11^C]ERGO PET radioligand as a robust marker for neuroinflammation in AD. In this study, after PET imaging, animals (non-treated 5XFAD (*n* = 3) and ERGO-treated 5XFAD mice (*n* = 5)) were perfused, and the brain sections were stained with antibodies against GFAP and IBA1 for neuroinflammatory markers, astrocytes and microglia, respectively. Data in Figure 4H–J showed that the level of GFAP-positive astrocytes in the hippocampus was reduced by nearly 50% in the treated group (Figure 4I) compared to non-treated 5XFAD mice (Figure 4H). Similarly, treating mice with ERGO resulted in a nearly 80% reduction in IBA1-positive microglia (Figure 4K (non-treated 5XFAD)**,** L (treated 5XFAD), M), *p* = 0.0007.

### 2.7. ERGO Treatment Rescues Glucose Metabolism in 5XFAD Mice

The brain consumes a significant amount of glucose, approximately 25% of the body’s glucose [39]. It is well known that glucose metabolism is diminished in the early onset of AD. Therefore, [^18^F]FDG PET imaging to assess glucose levels could be used for AD diagnosis. For this purpose, three cohorts of age-matched mice (6-month-old, *n* = 3 for each cohort), including WT, non-treated 5XFAD, and ERGO-treated 5XFAD (six doses, each 50 mg/Kg during the course of 2 weeks), were studied. Afterward, the animals were injected with [^18^F]FDG probe (15 MBq, 100 μL/mouse) and underwent 20 min dynamic PET scans and CT scans. The data in Figure 5 showed that non-treated 5XFAD mice (D–J) had lower glucose metabolism compared to WT mice (A–C,J). In stark contrast, glucose levels were significantly elevated (*p* < 0.05) in the brains of ERGO-treated compared to non-treated 5XFAD mice (Figure 5G–J), suggesting the rescuing role of ERGO as an antioxidant in glucose metabolism in AD.

## 3. Discussion

There is convincing evidence demonstrating that free radical damage and oxidative stress play a pivotal role in the early onset of AD [40]. Postmortem brain tissues from AD patients have significant extent of oxidative damage associated with extracellular Aβ plaques and intracellular neurofibrillary tangles [41]. Further, metals have been described to be involved in different pathophysiological mechanisms associated with neurodegenerative diseases, including AD [42]. With the availability of magnetic resonance imaging, iron has been mapped in the brain regions usually associated with Aβ deposits in AD [43,44]. Iron accumulation in the Aβ plaques has also been detected by MRI [45]. Another factor that makes the brain more susceptible to oxidation in AD is the presence of transition metals [46], such as Fe, Zn, and Cu [47,48]. These metals serve as a catalyst for the redox-generated free radicals, such as hydroxyl radicals, which have been proposed to initiate Aβ aggregation [12]. Free metal ions have been detected with abnormally high concentration in the ageing brain, as well as during several neurodegenerative disorders [11,49,50,51]. In AD brains, endogenous metals, such as Fe, Zn, and Cu, are found at a higher concentration in Aβ plaques compared to healthy brains [52]. For example, high levels of Cu (400 μM) and Zn (1 mM) were reported in AD brain compared to 70 μM and 350 μM, respectively, in healthy brain [11,53].

Taking all these factors into account, it is apparent that reducing the levels of ROS and/or neutralizing transitions metals, which are catalysts for ROS production, would benefit the brain. ROS- and transition metal-chelating therapy [54,55] have been tested in clinical trials for the treatment of AD using antioxidants and chelators, respectively. These trials show that dietary uptake of antioxidant supplements, such as vitamin E or both vitamins C and E, may lower the risk of AD [56,57]. Likewise, clinical trials using clioquinol derivatives as metal chelators showed reduced cerebrospinal fluid (CSF) Aβ-42 concentration [58]. Despite some initial success, there have been several issues associated with such antioxidants and chelators, which have become the subject of debate, including toxicity and the overall limited ability to cross the blood–brain barrier (BBB) [59]. In that regard, ERGO has more advantage, given it is a dual ROS/metal scavenger, which can be distributed to the brain independent of the BBB. In this project, we show that treating 5XFAD mice with high doses of ERGO resulted in reduced Aβ burden, oxidative stress, and rescued glucose metabolism, and with some modest improved cognition. Our data are supported by the past observation that ERGO could diminish oxidative stress when it is elevated with high concentrations [60].

This study shows that imaging biomarkers associated with AD pathogenesis may offer a distinct opportunity for monitoring the response to therapy. Interestingly, we observed both cognitive and molecular benefits of ERGO treatment in 4-month-old 5XFAD mice prior to the onset of significant behavioral deficits and at least 2 weeks after ERGO treatment had been stopped. This suggests either a lasting benefit of the compound or at least a delay in onset of genotype-associated behavioral and molecular changes. Importantly, ERGO did not have any detrimental impact on activity levels, anxiety-like behaviors, or motor learning, which would limit its utility as a therapeutic intervention. All three groups of mice were able to learn the cue–context–shock associations, as evidenced by increased freezing behaviors during the two test trials. However, learning was weaker in the non-treated 5XFAD mice, which appeared to freeze at approximately half of the rate of WT and ERGO-treated 5XFAD animals during the presentation of the cue (tone) in the novel context. This task requires functional connections between the hippocampus, frontal cortex, cingulate cortex, as well as the amygdala, which may be particularly important in cue retrieval, which was preserved by ERGO treatment [61]. Context retrieval is thought to be more directly dependent on hippocampal inputs. While we did not demonstrate a role for ERGO treatment in context retrieval, it should be noted that, at this age, there were also no deficits in the non-treated 5XFAD mice in this portion of the task. Similarly, we detected no clear deficits in the 5XFAD mice on the novel object recognition task, which is also thought to be highly dependent on intact hippocampal function. However, we used a protocol in which the exposure and test trials were separated by approximately 2 min. It is possible that had we imposed a longer delay that recall would have been impaired in the 5XFAD mice, which would then have allowed a clearer evaluation of any potential beneficial effects of the ERGO treatment. It will therefore be important to further assess the potential impacts of short- or longer-term ERGO treatment in more cognitively demanding tasks or in older animals in which cognitive deficits are more widespread. One potential future study would focus on treating young mice with ERGO and let them mature to old age before assessing their cognition. However, we may encounter potential dilemmas when using old AD subjects in this preventive approach. The intervention at the late stage could abolish the Aβ burden or alleviate oxidative stress, but that may not translate into improved cognition due to tissue atrophy.

Regardless of the modest cognition data on treated young mice, this work demonstrated the power of imaging technology for assessing the response during therapy. Since AD is a complicated disease, it seems likely in vivo tracking of a combination of different biomarkers would provide adequate information about the progress of the disease.

## 4. Materials and Methods

### 4.1. ERGO Formulation

L-ERGO was purchased from Cayman Chemical (Ann Arbor, MI, USA, catalog number: 14905) and used without any further purification, albeit the material was authenticated and confirmed via mass spectrometry and spectrophotometry before use. The compound was diluted in dd. water at room temperature at a concentration of 25 and 50 mg/Kg as a fresh stock solution for every gavage treatment. This dose range was higher than the dose used in a clinical trial [62], where adults consumed approximately 40 mg of ERGO. Based on the past report regarding the efficacy of ERGO at elevated concentrations [60], we wanted to test the effectiveness of ERGO in alleviating oxidative stress at high doses.

### 4.2. Mass Spectrometry Analysis

Liquid chromatography–mass spectrometry (LC–MS) was performed on the sample using a Waters Aquity HPLC and a Thermo Fisher LTQ-Orbitrap XL. The MS was operated in an FT mode in positive ion mode, with 60 k resolution scanning between 200 and 2000. About 10 μL of the sample was injected onto a symmetry 300 C18 3.5 μm, 2.1 × 100 mm column, and mobile phases (A) 0.2% formic acid and 0.05% TFA in water/acetonitrile (9/1) and (B) 0.2% and 0.05% TFA in acetonitrile/water/isopropanol (4/1/5) with a flow rate of 300 μL/min. The gradient held steady at 100% phase A for 1 min before changing to 100% phase B at 10 min and holding for 1 min before returning to the starting conditions and re-equilibrating for 3 min. The data collected were examined using Qual Browser software 2.0.7 SP1 (Thermo Fisher Scientific, Waltham, MA, USA).

### 4.3. ROS Measurement

On day 1, HeLa cells or SH-SY5Y cells were seeded into a 96-well plate with a density of 10,000 cells/mL. On day 2, the DCFH-DA probe (Cell Biolabs, Inc., San Diego, CA, USA) was formulated in cell culture media (Gibco Dulbecco’s Modified Eagle Medium (DMEM) and dispensed to the allocated wells at a final concentration of 1×, as recommended by the manufacturer. The plate was then incubated in the dark for 1 h at 37 °C before excess probe was removed, and the cells were washed repeatedly 3 times with Dulbecco’s Phosphate-Buffered Saline (DPBS). Then, the ERGO solution was added to the designated wells at concentrations ranging from 5 to 100 mM. The free radical initiator solution was prepared at a concentration of 1×, as per manufacturer’s recommendation, and added to all wells. Immediately afterward, fluorescence measurement began using a microplate reader (Biotek Industries, Agilent Technologies, Winooski, VT, USA) with an excitation and emission wavelength of 485 nm and 528 nm, respectively. Each ERGO concentration was measured in pentaplicate.

### 4.4. Metal Scavenging Assays

#### 4.4.1. Nickel Assay

ERGO’s metal-binding affinity was tested via a 2D-HPLC equipped with a nickel-based HisTrap HP column (Cytiva, Marlborough, MA, USA). The 2D-HPLC was performed with a Hitachi LaChrom Elite L-2455 Diode Array Detector linked to a Hitachi L-2130 pump (Hitachi High-Technologies, Corp., Tokyo, Japan). Two reservoirs of the mobile phase, including PBS and PBS with 50 mM imidazole, were programed so that the first 15 min of the run would be covered with 100% PBS, followed by switching to 100% of the eluting buffer, which is composed of PBS with 50 mM imidazole. In a typical experiment, histidine-containing camelid antibodies (250 ng, MW 14 KDa) dissolved in 200 µL PBS were first injected into the HPLC system. In this setup, the histidine-containing camelid antibodies were immobilized in the nickel column exclusively during the first 15 min with PBS buffer. However, when the mobile phase was changed to PBS with a large concentration of imidazole, which would compete and displace His-tag from camelid antibodies, it resulted in antibody elution. In another experiment, the antibody was injected as described above but with the addition of ERGO (50 nM). Right at the beginning of the run, His-tag camelid antibodies were eluted, indicating that ERGO competes with the His tag for nickel, resulting in an early washout of the antibodies. All analyses were performed via the EZChrom Elite software package (Agilent Technologies, Winooski, VT, USA).

#### 4.4.2. Ferrous Ion Chelating (FIC) Assay

The iron chelating activity of ERGO was determined using a FIC assay, which was obtained from Zen-Bio Inc. (Durham, NC, USA). The assay was performed on a 96-well plate with triplicated samples, standards, background, and max absorbance wells. All of the assay absorbance values were subtracted from the average background absorbance, comprising water and assay buffer (1:1). Maximal absorbance was measured by averaging the absorbance values of 3 wells containing only the FeSO4 and assay buffer. For the testing wells, each one contained the FeSO4 (50 μL) and ERGO samples at different concentrations, including 0, 5, 25, 50, and 100 mM. Control wells contained EDTA standards and a solution of FeSO_4_ (50 μL), except for the background wells. The assays were started by adding ferrozine solution (100 μL, 1×) to each well using a multichannel pipet, with the final volume of each well being 200 μL. The assaying plate was incubated at room temperature for 10 min; then, the plate was read at 562 nm using a Biotek Synergy 4 plate reader. Absorbances were averaged, and the background absorbance was subtracted from the sample and maximum averages (Abs_test_ and Abs_max_). The ferrous iron chelating percentage was then calculated using the following equation: Ferrous ion chelating (%) = 100 × (Abs_max_ − Abs_test_)/Abs_max_.

#### 4.4.3. Cupric Ion Chelating (CIC) Assay

The copper chelating activity of ERGO was determined using a CIC assay (Zen-Bio Inc., Durham, NC, USA). The assays contained triplicated samples, standards, background, and max absorbance wells. All the assay absorbance values were subtracted from the average background absorbance, comprising water and assay buffer (1:1). Maximal absorbance was measured by averaging the absorbance values of 3 wells containing only the CuSO_4_ and assay buffer. For the testing wells, each one contained CuSO4 (30 μL) and ERGO samples at different concentrations, including 0, 16, 31, 62, 125, 250, 500, and 1000 mM. Control wells contained EDTA standards and a solution of CuSO_4_ (30 μL), except for the background wells. The plate was incubated at room temperature for 5 min. The assays were started by adding pyrocatechol violet solution (8.5 μL, 1×) to each well, with the final volume of each well being 268.5 μL. The assaying plate was incubated at room temperature on a shaker for 10 min, followed by an additional 10 min incubation at room temperature without shaking. The plate was then read at 632 nm using a Biotek Synergy 4 plate reader. Absorbances were averaged, and the background absorbance was subtracted from the sample and maximum averages (Abs_test_ and Abs_max_). The cupric ion chelating percentage was then calculated using the following equation: Cupric ion chelating (%) = 100 x (Abs_max_ − Abs_test_)/Abs_max_.

### 4.5. Animals

A colony of 5XFAD mice obtained from Jackson Laboratories was maintained by crossing with WT C57BL/6J, as we reported in the past [63]. The animals were genotyped by polymerase chain reaction (PCR) using DNA obtained from the tail or ear tissue samples. After PCR amplification, the DNA product was analyzed using a 1% agarose gel, amyloid precursor protein (APP) transgene = 377 bp, and presenilin 1 (PSEN1) transgene = 608 bp. 5XFAD mice were maintained as heterozygous. The animals were treated with an ERGO formula beginning at the age of 2 months. Animal experiments were conducted in accordance with the guidelines established by the Vanderbilt University’s Institutional Animal Care and Use Committee (IACUC) and the Division of Animal Care and approved by Vanderbilt IACUC, protocol number M1700044.

#### Gavage Treatment

Mice (2-month-old) were dosed (50 mg/Kg, less than 100 μL) by means of oral gavage. The procedure involves passing a reusable oral gavage needle through the mouth and placing it atop the esophagus of an awake animal in the way to encourage the animal to swallow the formulation voluntarily. The curvature of the syringe along with the extra smooth round ball stainless-steel tip ensured minimal discomfort to the treated animals. Since the gavage procedure involved restraining of the animal, it might cause stress [64], which is a potential confounding experimental endpoint in behavioral assessments. To ameliorate the potential discrepancy between the treated and non-treated cohorts, the latter underwent the same oral gavage process but were given only the vehicle, sterilized water (100 μL).

### 4.6. Behavioral Experiments

The primary objective of this work was to answer whether ERGO treatment benefits the brain and improves cognition. At the end of the ERGO-based therapy, animals were transferred to the Vanderbilt Neurobehavioral Laboratory to acclimate for at least 3 days before testing.

#### 4.6.1. Elevated Zero Maze (EZM)

We recorded the exploratory activity in open versus closed zones of a standard EZM during a 5 min trial. At the start of the trial, mice were placed in an open zone of the maze and allowed to explore freely while being videotaped from above. The floor of the maze was 5 cm wide. Closed zones had walls of approximately 30 cm height and light levels of 215–280 lux depending on where they were measured. Open zones had a small lip of ~0.5 cm and light levels of 349–469 lux. Trials were observed by an experimenter in an adjacent room.

All behavioral apparatus were cleaned with 10% ethanol solution between trials to sanitize the equipment and minimize the odor trails lefts by previous animals. The animal’s location within the maze and distance traveled were analyzed using AnyMaze (Stoelting Co., Wheat Lane Wood Dale, IL, USA).

#### 4.6.2. Locomotor Activity

Exploratory locomotor activity was measured in specially designed chambers measuring 27 × 27 cm (Med Associates), housed in sound-attenuating cases over a 30 min period. Horizontal and vertical activities within the chambers were automatically recorded via the breaking of infrared beams.

#### 4.6.3. Rotarod

Neuromuscular ability and motor learning were assessed using a standard rotarod (Ugo Basile). Mice were placed on a 6 cm wide section of a ridged rod that rotated slowly. The rod began rotating at 4 rpm and ramped up to a maximum speed of 40 rpm by 4 min (total test time 5 min, max speed for final minute). Mice were allowed to complete 3 trials per day on 3 consecutive days. Time to fall was recorded automatically when mice fell from the rod onto a base plate.

#### 4.6.4. Novel Object Recognition

Mice were first allowed to habituate to an open arena for 5 min (white acrylic box, approximately 40 cm^2^) located under a camera to record the position and movement during trials. Immediately following this acclimatization phase, each mouse was removed from the arena, which was cleaned with 10% ethanol, and two identical objects were placed in the arena in the center of each arena half. Mice were returned to the arena and allowed to freely explore the objects for 6 min. Mice were then once again removed from the arena, which was again cleaned with 10% ethanol. A third identical exemplar of the familiar object was then added to the arena with one novel object. The position of the novel object was balanced across groups. Exploration of the two objects was permitted for 6 min until the trial was terminated. Preference for either object was inferred from exploratory proximity, which was recorded automatically using AnyMaze using a target area comprising a circle with diameter 2 cm larger than the target objects. A recognition index was calculated to assess preference for the novel versus the familiar object as time in proximity to the novel object (T_N_) divided by time spent in proximity to either novel or familiar (T_F_) objects (T_N_/(T_N_ + T_F_)).

#### 4.6.5. Fear Conditioning

Mice were placed in a sound-attenuating chamber with a wire grid floor capable of transmitting an electric shock. All movements were monitored by cameras fixed to the inside of the doors. Training trials were 8 min long, during which time a 30 s tone was played three times, which co-terminated with a small shock delivered through the metal floor (1 s, 0.5 mA). Following training, mice were transferred to a holding cage in a second ante-chamber and were not reintroduced to (naïve) cage mates until all the mice had undergone training trials. Twenty-four hours following the training trial, the mice were exposed to the same chamber to assess memory for the testing context (4 min trial, no tone, no shock). Mice were tested in the same chambers, in the same test order, and under the same lighting and other experimental conditions as training. One hour following the context retrieval trial, the mice were tested again in a novel context for which a second identical chamber was used, but it was altered by using a white plexiglass wall and floor insert and a 10% vanilla scent placed in a weigh dish within the outer chamber (not accessible to the mouse). During this 4 min trial, the previously exposed tone was played during the final 2 min, but there was no shock administered. Freezing behavior was monitored automatically.

### 4.7. Dynamic PET Imaging

The dynamic acquisition was divided into twelve 5 s frames, four 60 s frames, five 120 s frames, three 5 min frames, and six 10 min scans. The data from all possible lines of response (LOR) were saved in the list mode raw data format. The raw data were then binned into 3D sinograms with a span of 3 and ring difference of 47. The images were reconstructed into transaxial slices (128 × 128 × 159) with voxel sizes of 0.0815 × 0.0815 × 0.0796 cm^3^, using the MAP algorithm with 16 subsets, 4 iterations, and a beta of 0.0468. For anatomical co-registration, immediately following the PET scans, the mice received a CT scan in a NanoSPECT/CT (Mediso, Washington DC) at an X-ray beam intensity of 90 mAs and X-ray peak voltage of 45 kVp. The CT images were reconstructed into 170 × 170 × 186 voxels at a voxel size of 0.4 × 0.4 × 0.4 mm^3^. The PET/CT images were uploaded into Amide software (www.sourceforge.com, accessed on 1 May 2022), co-registered to an MRI template made in-house, and volumetric regions of interest were drawn around the cortex, hippocampus, striatum, thalamus, and cerebellum, in addition to the whole brain. The PET images were normalized to the injected dose, and the time-activity curves (TACs) of the mean activity within the ROIs were estimated for the entire duration of the scans.

### 4.8. Immunohistochemistry (IHC)

Brains embedded in OCT were cut into sagittal sections (10 µm) using a Tissue-Tek cryostat and mounted onto charged glass slides. Prior to staining, the slides were washed with PBS (10 min); then, they were treated with blocking buffer (5% normal goat serum, 0.2% Triton X-100, 0.5% bovine albumin in PBS) for 1 h at room temperature. The treated sections were then incubated overnight at 4 °C with primary anti-GFAP antibody (1:100 dilution, Biolegend San Diego, CA, USA, catalog number: 644701). Slides were washed with PBS (3×) for 10 min each; the sections were subsequently incubated with secondary antibody goat anti-mouse Alexa Fluor 488 (1:200 dilution, Thermo Fisher Scientific, Carlsbad, CA, USA, catalog number: A-11001) for 30 min at room temperature. The sections were then washed with PBS twice for 10 min and once for 30 min, and coverslipped with an antifade mounting medium (Vector Laboratories, Burlingame, CA, USA, catalog number: H-1200-10) before observation under a fluorescence microscope.

### 4.9. Data Analysis

Quantitative analysis of the PET imaging and IHC data was performed using imageJ software. The data were imported to GraphPad Prism version 9 for Mac (GraphPad Software, San Diego, CA, USA) for statistical analysis. Differences between groups were tested using an unpaired *t*-test.

Behavioral data were analyzed using Prism 9 for Mac OS. Single outcome measures for elevated zero maze (EZM) and locomotor activity chambers were analyzed using univariate ANOVA with Tukey’s multiple comparisons post hoc tests, following significant omnibus ANOVA. Fear conditioning data were analyzed using non-parametric approaches because the data were not normally distributed (Brown–Forsythe test *p* < 0.05). We therefore used the Kruskal–Wallis test for single dependent variable (freezing in familiar context). Data were first assessed for significant effects of sex on all outcomes. Since there were no differences observed, the data for male and female animals were combined.

Data are given as mean ± standard error of the mean (SEM). Different levels of significance were described as * *p* < 0.05, ** *p* < 0.01, and *** *p* < 0.0001.

## 5. Conclusions

The data obtained from this work demonstrated the potential use of ERGO for treating AD. It is worth noting that the concentration of ERGO used in this therapy and the assay for scavenging ROS was relatively high. Since ERGO is a natural product, it does not have therapeutic efficacy on the same par as pharmaceutical drugs. There are two approaches we envision for future applications. One of those would be focusing on prevention. With a high concentration of ERGO in dietary sources, such as in mushrooms, it is anticipated that the consumption of mushroom, in the long term, benefits the brain, and it could be helpful for the prevention of AD. Another approach perhaps focuses on converting this natural antioxidant into therapeutic drugs. A general structure–activity relationship (SAR) study conducted by modifying the chemical genetics surrounding ERGO could lead to a potent agent with improved efficacy.

## Figures and Tables

**Figure 1 pharmaceuticals-15-00742-f001:**
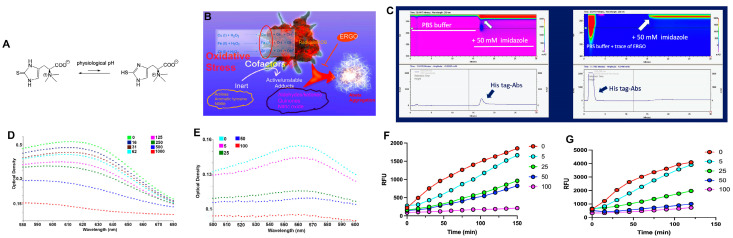
(**A**) The chemical structures of ERGO and the tautomerized isoforms at physiological pH; (**B**) The hypothesis is that ROS and transition metals in oxidative stress environments could be involved in the initiation of Aβ aggregation; with the capability to scavenge both ROS and metals, ERGO could prevent this event; (**C**) The 2D-HPLC was equipped with a nickel column to demonstrate the strong affinity of ERGO for metals. Upon exposure to the nickel column, His tag-Abs would be immobilized onto the column due to the strong binding of His tag (6 histidines) for the metal. The His tag-Abs can be eluted from the column using a buffer with a high concentration of imidazole (left figure). However, in the presence of ERGO (14.5 mM), the His tag-Abs were washed out immediately (right figure); (**D**,**E**) ERGO scavenges copper and iron, respectively. In these assays, Cu(II) or Fe(II) form a complex with pyrocatechol violet (PV) dye or ferrozine, resulting in a chromophore with strong absorbance at the wavelengths of 632 nm (copper) or 562 nm (iron). However, in the presence of ERGO, these transition metals were scavenged, leading to decreased PV–metal and ferrozine–metal complex concentrations, which resulted in a loss of absorbance at respective wavelengths. (**F**,**G**) In this cell-based assay, Hela (**F**) or neuroblastoma SH-SY5Y cells (**G**) were treated with DCFH-DA dye, which will diffuse into the cytoplasm and be trapped there after deacetylation caused by cellular esterases. The resulting nonfluorescent dye was oxidized by free radicals, emitting a robust fluorescent signal (red curve). ERGO scavenges free radicals in the cells and leads to fluorescence attenuation. The fluorescence quenching is concentration dependent. At a dose of 100 mM, ERGO effectively quenches the fluorescence totally. Each point is the average of a pentaplicate assay. *p* < 0.0001.

**Figure 2 pharmaceuticals-15-00742-f002:**
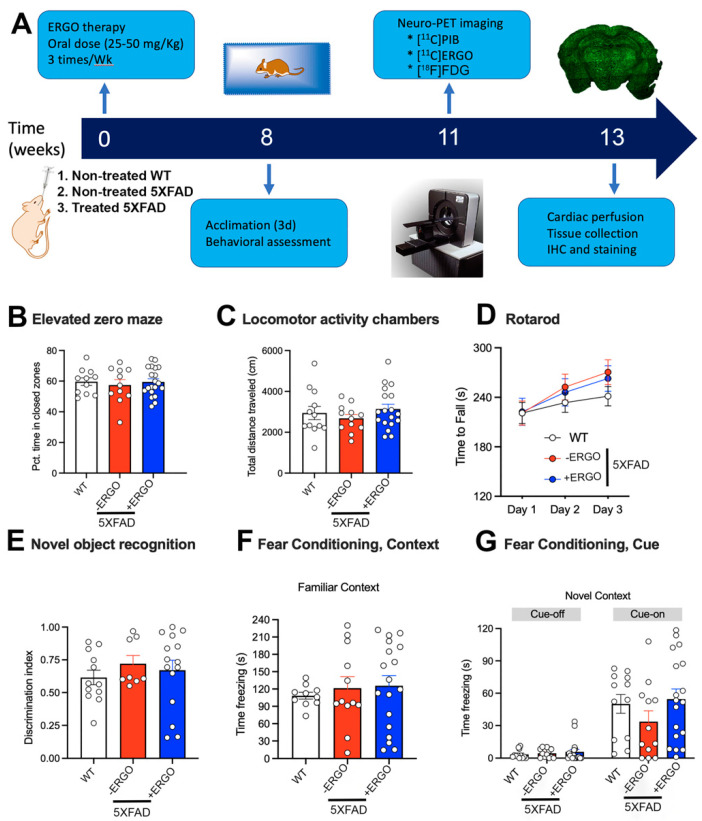
Timeline of the therapy. Three independent cohorts of animals were treated with either water as vehicle, denoted as “non-treated” or ERGO ranging from 25 to 50 mg/Kg via gavage three times per week for 8 weeks, with mice from each group represented in each cohort. Afterward, the animals underwent behavioral testing and PET imaging to assess the molecular biomarkers related to AD. Finally, the animals were sacrificed, and brain sections were prepared for immunohistochemical analysis. (**A**) Schematic of experimental timeline; (**B**) Elevated zero maze, percent time spent in closed zones during 5 min trial; (**C**) Exploratory locomotor activity, total distance traveled across a 30 min trial; (**D**) Rotarod, latency to fall across 3 days of testing, mean from three daily trials; (**E**) Novel object recognition, discrimination index (T_Novel_/(T_Novel_ + T_Familiar_)); (**F**,**G**) Conditioned fear task, (**F**) Context-based recall, time spent freezing across 4 min trial; (**G**) Cue-based recall, time spent freezing during “cue off” (2 min) and “cue-on” (2 min) portions of the trial. Wild-type control *n* = 6 male, *n* = 6 female, non-treated 5XFAD *n* = 6 male, *n* = 6 female, ERGO-treated 5XFAD *n* = 9 male, *n* = 9 female; (**D**) *p* < 0.001, Main effect of training day; (**G**) * *p* < 0.05, Cue-on freezing time compared to cue-off freezing time for each group.

**Figure 3 pharmaceuticals-15-00742-f003:**
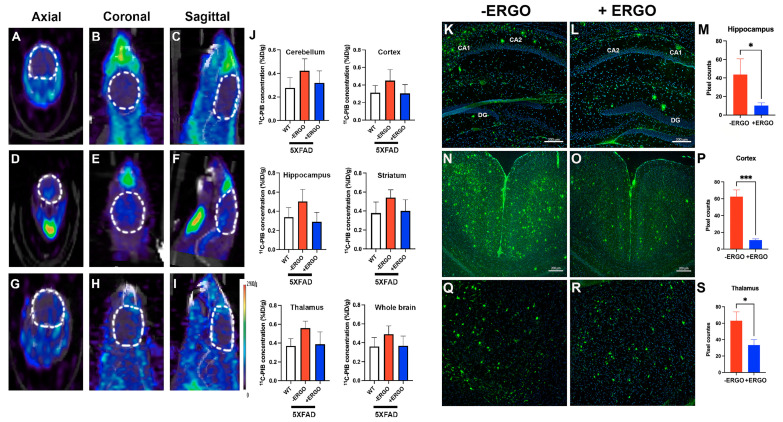
Reduced Aβ expression in the brains of ERGO-treated 5XFAD mice. After the therapy, all cohorts of animals were screened non-invasively using PET/CT scans to assess the levels of Aβ using [^11^C]PIB PET radioligand. The background PET signal was first established using WT mice (**A**–**C**) (*n* = 4). A similar process was performed on non-treated 5XFAD mice, and the data demonstrated significant uptake and retention of [^11^C]PIB, reflecting the enhanced expression of Aβ (**D**–**F**) (*n* = 5). Meanwhile, PET signal was much lower, suggesting less Aβ loads in the treated 5XFAD mice (**G**–**I**) (*n* = 7). Semi-quantitative analysis of the in vivo uptake of [^11^C]PIB PET radioligand in different subregions of the brains of three cohorts of mice (**J**); *p* < 0.05 for each subgroup. The in vivo PET data corroborate Aβ immunohistochemistry (green, 488 nm channel), as shown in representative staining of Aβ on DAPI (blue)-stained coronal brain slices (10-μm) using 6E10 antibodies of ERGO-treated versus non-treated 5XFAD mice (*n* = 3, each cohort, three slides per mouse) and subsequent quantitative analysis of the pixel counts after thresholding for the hippocampus (**K**–**M**), pyramidal cortex (**N**–**P**), and thalamus (**Q**–**S**). * *p* < 0.05, *** *p* = 0.0004.

**Figure 4 pharmaceuticals-15-00742-f004:**
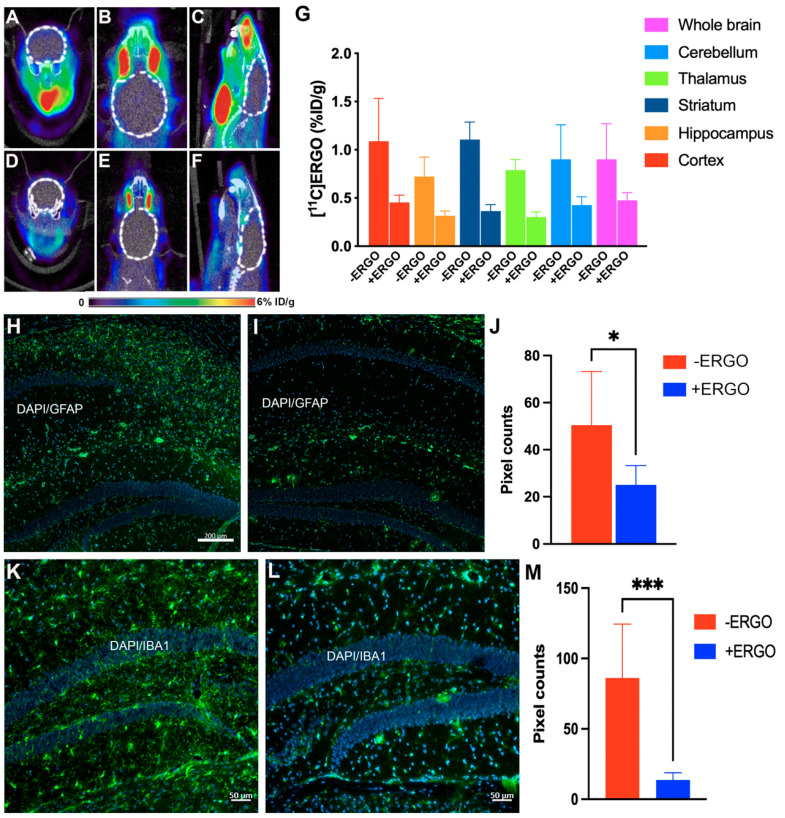
Attenuation of oxidative stress in ERGO-treated 5XFAD mice using the [^11^C]ERGO radioligand. Representative of in vivo PET imaging data to assess oxidative stress of non-treated 5XFAD (*n* = 3) (**A**–**C**) versus ERGO-treated 5XFAD (*n* = 5) (**D**–**F**). The axial (**A**,**D**), coronal (**B**,**E**), and sagittal (**C**,**F**) images were obtained from the PET/CT scans. Semi-quantitative analysis of the in vivo uptake of [^11^C]ERGO PET radioligand in different subregions of the brains of two cohorts of 5XFAD mice (**G**); the difference between treated and non-treated cohorts is quantified with *p* < 0.05 for all subregions of the brain. Representative of immunohistochemical staining and the corresponding quantification data of GFAP-positive astrocytes (**H**–**J**) (green, 488 nm channel) and IBA1-positive activated microglia (**K**–**M**) (green, 488 nm channel) on coronally DAPI-stained (blue) brain slices (10 μm) of non-treated 5XFAD (**H**,**K**) versus ERGO-treated counterpart (I,L) (*n* = 5, each), * *p* < 0.05, *** *p* = 0.0007.

**Figure 5 pharmaceuticals-15-00742-f005:**
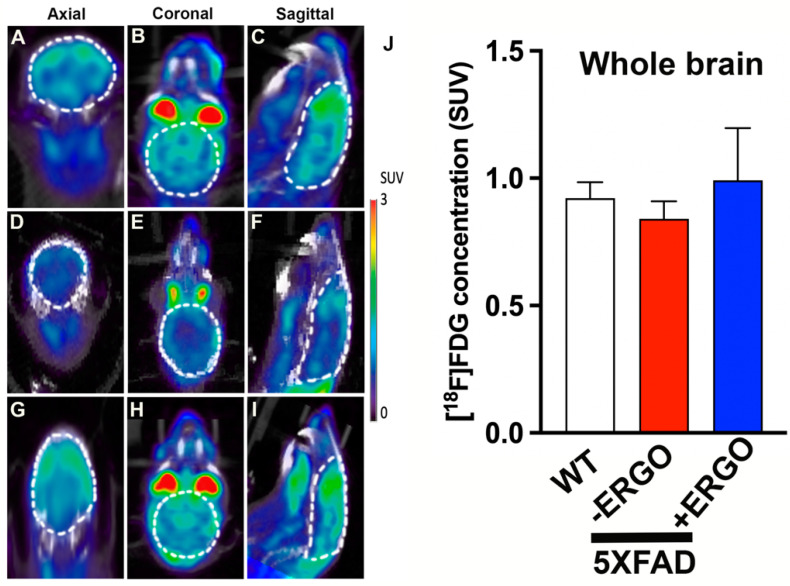
Imaging glucose metabolism using [^18^F]FDG PET radioligand. Representative of 20 min dynamic PET scans with focus on the brains of WT mice (**A**–**C**), non-treated 5XFAD (**D**–**F**), and ERGO-treated 5XFAD (**G**–**I**). The uptake of the [^18^F]FDG PET probe was quantified as standard uptake values (SUV) in the brains, and the SUVs were compared between ERGO-treated and non-treated 5XFAD (*n* = 3, each) at *p* < 0.05 (**J**).

## Data Availability

Data is contained within the article or Appendix A.

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
