# Peer review of "Longitudinal Consumption of Ergothioneine Reduces Oxidative Stress and Amyloid Plaques and Restores Glucose Metabolism in the 5XFAD Mouse Model of Alzheimer’s Disease"

_pharmaceuticals, 2022, doi:10.3390/ph15060742_

Round 1

Reviewer 1 Report

  1. Specify in the abstract if the modest improvement is statistically significant or not.
  2. In the abstract indicate the age of the young AD mice.
  3. Add to the methods the procedures for stereological analyses.
  4. Add to the discussion a section on the limitations of the study regarding not observing animals chronically.

Author Response

  1. Specify in the abstract if the modest improvement is statistically significant or not.

Answer: The reason we mentioned modest improvement in cognition of treated mice is that not all studies show a significant difference between cohorts. Thus, it is very difficult to show all statical values in the abstract for a general statement. We hope the readers can see all the detailed statistical values for each individual test in section 2.4. ERGO treatment prevents early cognitive deficits in 5XFAD mice.

  1. In the abstract indicate the age of the young AD mice.

Answer: We added this information in the abstract.

  1. Add to the methods the procedures for stereological analyses.

Answer: We described stereological analysis of the data in the Supplementary data. And we describe the immunohistochemistry procedure meticulously in section 4.9. Immunohistochemistry.

  1. Add to the discussion a section on the limitations of the study regarding not observing animals chronically.

Answer: We added that statement at the end of the Discussion.

Reviewer 2 Report

The manuscript entitled “Longitudinal consumption of ergothioneine reduces oxidative stress, amyloid plaques, and improves cognition in the 5XFAD mouse model of Alzheimer’s disease” addresses the beneficial effects of ergothioneine, antioxidant agent and cytoprotectant, on the molecular events linked to early Alzheimer pathogenesis in 5XFAD mice model. Strikingly, young 5XFAD mice did not show a cognitive deficit. However, ergothioneine treatment to the transgenic mice lowered amyloid plaque deposition, and oxidative stress markers, and rescued glucose metabolism.

Comments:     

1) Since young 5XFAD mice did not show a significant cognitive deficit as revealed by the behavioral tests, authors are advised to remove “improve cognition” and modify the title of the current study, for example, to: “Longitudinal consumption of ergothioneine reduces oxidative stress and amyloid plaques in the 5XFAD mouse model of Alzheimer’s disease”.

2) The authors are advised to outline in the introduction section the novelty of the present work and how is the study is different from previous literature that has already described the effect of ergothioneine on the deposition of amyloid-beta fibrils, such as:

  1. A) Cheah et al., 2019 (Inhibition of amyloid-induced toxicity by ergothioneinein a transgenic Caenorhabditis elegans model, FEBS Lett. 2019 Aug;593(16):2139-2150. doi: 10.1002/1873-3468.13497). 
  2. B) Song et al., 2014 (Ergothioneine and melatonin attenuate oxidative stress and protect against learning and memory deficits in C57BL/6J mice treated with D-galactose, Free Radic Res. 2014 Sep;48(9):1049-60. doi: 10.3109/10715762.2014.920954). 

3) In the present study design, there is an issue with using young 5XFAD mice which are assumably representing Alzheimer’s disease. Yet, all the used behavioral tests revealed non-significant changes in cognition which raises concern about the validity of the present model. Should the authors have used old 5XFAD mice to show significant differences versus the WT animals? This point needs to be discussed in the discussion section.

-  How did the authors decide about the age of animals (2 months)? Why did not the authors use 8-month and older 5XFAd mice which are expected to demonstrate significant differences in the behavioral tests vs. WT mice?

4) How did the authors decide about the doses of ergothioneine in mice (50 mg/kg/day)? How is the dose relevant to the human dose using the Human effective dose (HED) formula= animal dose x animal Km/ human Km (Nair AB, Jacob S. A simple practice guide for dose conversion between animals and humans. J Basic Clin Pharm. 2016 Mar;7(2):27-31). Please also provide proper citations for selecting such a dose. Authors are advised to carefully address this point and add the answers to the comment in the material and methods section.

5) How did the authors decide about the length of treatment with ergothioneine (3 times per week for 4 weeks only? How is the length of treatment relevant to that of the human being? Please, provide proper citations.

6) What is the LD50 for ergothioneine in mice? Is the used dose safe?

7) Since the oral gavage treatment triggers stress in animals which should is a confounding factor in the used behavioral tests, why did not the authors use ergothioneine in drinking water, particularly, since its solubility in water is high?

8) In section 3.5 (Animals), please specify the gender of the used 5XFAD mice (male and female). Why did not the authors choose to use male mice only, for example. Were males segregated from female mice to avoid pregnancy, please, elaborate on these issues in section 3.5.  

9) The legend of figure 2 states that “Three independent cohorts of animals were treated with ERGO ranging from 25-50 mg/Kg via gavage 3 times per week for 4 weeks with mice from each group represented in each cohort”.

First, why a range of doses was used? why did not the authors use a fixed-dose 50mg/kg/day, for example?

Second, the statement is confusing since only one group should be receiving ergothioneine which is 5XFAD+ERGO. Please, rewrite the sentence to clarify this issue.

10) The authors are advised to provide in the ethics statement (Institutional Review Board Statement) the ethical approval number.

11) In line 524, the authors state that “Mice (2-month-old) were dosed (50 mg/Kg, less than 100 microliters) by means of oral gavage”. Please, specify whether the 100 microliters was used per mouse or per Kg?

12) The authors are advised to add the catalog number for ergothioneine and the used antibodies.

Author Response

Reviewer 2

  1. Since young 5XFAD mice did not show a significant cognitive deficit as revealed by the behavioral tests, authors are advised to remove “improve cognition” and modify the title of the current study, for example, to: “Longitudinal consumption of ergothioneine reduces oxidative

stress and amyloid plaques in the 5XFAD mouse model of Alzheimer’s disease”.

Answer: We modified the title as suggested by the reviewer.

  1. The authors are advised to outline in the introduction section the novelty of the present work and how is the study is different from previous literature that has already described the effect of ergothioneine on the deposition of amyloid-beta fibrils, such as:

  1. A) Cheah et al., 2019 (Inhibition of amyloid-induced toxicity by ergothioneinein a transgenic

Caenorhabditis elegans model, FEBS Lett. 2019 Aug;593(16):2139-2150. doi: 10.1002/1873-

3468.13497).

  1. B) Song et al., 2014 (Ergothioneine and melatonin attenuate oxidative stress and protect against

learning and memory deficits in C57BL/6J mice treated with D-galactose, Free Radic Res. 2014

Sep;48(9):1049-60. doi: 10.3109/10715762.2014.920954).

Answer: We thank the reviewer for this notion. We added the comments in the Introduction to highlight

the novelty of the work in relation to the past observations made by these two references.

  1. In the present study design, there is an issue with using young 5XFAD mice which are assumably representing Alzheimer’s disease. Yet, all the used behavioral tests revealed nonsignificant changes in cognition which raises concern about the validity of the present model.Should the authors have used old 5XFAD mice to show significant differences versus the WT

animals? This point needs to be discussed in the discussion section.

- How did the authors decide about the age of animals (2 months)? Why did not the authors use

8-month and older 5XFAd mice which are expected to demonstrate significant differences in the

behavioral tests vs. WT mice?

Answer: We discussed that issue and offered our view about testing old AD mice at the end of the Discussion. As we mentioned in the last rebuttal, this work focuses on the preventive effect of ERGO; we are not interested in testing old AD subjects in a prevention study in this manuscript. Based on a failed clinical study (N Engl J Med, 2014, 370, 322-333): anti-Abeta antibodies alleviate some Abeta

levels in the brains of AD patients but no reverse the course of the disease, nor did the treatment improve cognition because at this late stage, tissue atrophy already occurs is an irreversible process.

  1. How did the authors decide about the doses of ergothioneine in mice (50 mg/kg/day)? How is the dose relevant to the human dose using the Human effective dose (HED) formula= animal dose x animal Km/ human Km (Nair AB, Jacob S. A simple practice guide for dose conversion between animals and humans. J Basic Clin Pharm. 2016 Mar;7(2):27-31). Please also provide

proper citations for selecting such a dose. Authors are advised to carefully address this point and add the answers to the comment in the material and methods section.

Answer: We update this dose information in the Materials and Methods section 4.1. ERGO formulation. As we mentioned in the last rebuttal, this dose was extrapolated from the past clinical trial led by Prof. Beelman, an author of this manuscript. ERGO is not a toxic compound, and our objective is to escalate

the high dose to see if there is a response.

  1. How did the authors decide about the length of treatment with ergothioneine (3 times per week for 4 weeks only? How is the length of treatment relevant to that of the human being? Please, provide proper citations.

Answer: It is our oversight to mention 3 times/week for 4 weeks. As we stated in section 2.3, Timeline

of the therapy and processing: we treated mice 3 times a week over the course of 8 weeks. And the

treatment timeline in Figure 3A also confirms this information. For the animal study, this is a long and

intensive treatment. We are the first to develop this therapy; thus, no prior references are available.

  1. What is the LD50 for ergothioneine in mice? Is the used dose safe?

Answer:

The dose we use in this study is totally fine. We not only did not see toxicity or adversary effect on animals, but we also did not observe any detrimental impact on activity levels, anxiety-like

behaviors, or motor learning. This observation was discussed carefully in the

Discussion.

  1. Since the oral gavage treatment triggers stress in animals which should is a confounding factor in the used behavioral tests, why did not the authors use ergothioneine in drinking water, particularly, since its solubility in water is high?

Answer: There are many reasons why we need to treat ERGO via gavage. But here are two major issues: (i) If we treat animals via drinking, we can not guarantee animals receive the right dose since each individual drinks differently; (ii) ERGO is a costly compound, we can formulate a precise dose like a drug molecule, but we can not afford to make a drinking solution.

  1. In section 3.5 (Animals), please specify the gender of the used 5XFAD mice (male and female). Why did not the authors choose to use male mice only, for example. Were males segregated from female mice to avoid pregnancy, please, elaborate on these issues in section 3.5.

Answer: As we mentioned in the last rebuttal, it is very costly we segregate sexes in the study. No one can afford that in the proof-of-principle study. We breed animals in our lab, and what we care about is that the animals must be age-matched for the study. If we don’t breed, male and female mice should

stay in separate cages. That is a general procedure, which we think unnecessary to mention in the manuscript.

  1. The legend of figure 2 states that “Three independent cohorts of animals were treated with ERGO ranging from 25-50 mg/Kg via gavage 3 times per week for 4 weeks with mice from each group represented in each cohort”.

First, why a range of doses was used? why did not the authors use a fixed-dose 50mg/kg/day, for example? Second, the statement is confusing since only one group should be receiving ergothioneine

which is 5XFAD+ERGO. Please, rewrite the sentence to clarify this issue.

Answer: We explain the rationale for using the dose in section 4.1. ERGO formulation. And we also modify the statement in the legend to make it cleaer, and thanks the reviewer for this note.

  1. The authors are advised to provide in the ethics statement (Institutional Review Board

Statement) the ethical approval number.

Answer: This information is now updated in section 4.5 Animals.

  1. In line 524, the authors state that “Mice (2-month-old) were dosed (50 mg/Kg, less than 100 microliters) by means of oral gavage”. Please, specify whether the 100 microliters was used per mouse or per Kg?

Answer: We treated 100 μL per mouse.

  1. The authors are advised to add the catalog number for ergothioneine and the used antibodies.

Answer: We added this information in section 4.1. ERGO formulation and section 4.9. Immunohistochemistry

Reviewer 3 Report

In this work, Whitmore and colleagues evaluated the effect or ERGO on young 5XFAD mice.  They first assessed the compound’s anti-radical properties with different cell lines and Ni quelating properties before advancing to the in vivo studies.  Although the mice behaved similarly on several performance tests, their learning ability was limited.  This was further evaluated through direct (histochemistry) and indirect (PET) methods.  Overall I believe the manuscript presents solid data that supports the conclusions, it is well written and can be published soon on Pharmaceutics.  I have some recommendations to the authors to improve their manuscript before publications.  Please return these to the authors with a recommendation for minor revisions:

1. “Abeta” should be “Aβ” to match standard use of the name.  Please consider using the Greek letter for “beta”.

2. I believe Figure 1 can be better organized to improve the readers comprehensions.  Please consider increasing the graphics in size and indicate clearly to which ion they refer to.

3. Consider changing the information contained in phrases 188-190 to the materials and methods sections.

3.1. Consider changing the materials and methods section of place to right after the introduction.

4. In phrase 269-270 I suggest keeping “8 weeks” instead of “2 month-old”.

5. In figure 3, please present the immunohistochemistry images for the control brain slices.

6. I suggest improving image 4 by indicating which markers are labeled in the images for better readability (I believe such captions are standard and include the marker + counter-staining used; e.g. DAPI/GFAP or DAPI/IBA1).

7. Consider reducing figure 5 in size.

8. In phrase 382, the symbol for Iron was written twice.  Please address this.

9. In your study no controls of anti-ROS and/or metal chelators were use to understand fully the mechanism of ERGO on the brain.  Can you comment on this in your manuscript?

10. Please present the variation of weight for the mice populations used in this study.

11. Please clarify whether the mice had free access to food and water.

12. Please include the DNA analysis of the mice populations used (referred in section 3.5).

Author Response

  1. “Abeta” should be “Aβ” to match standard use of the name.  Please consider using the Greek letter for “beta”.

Answer: We updated this modification throughout the manuscript.

  1. I believe Figure 1 can be better organized to improve the readers comprehensions. Please consider increasing the graphics in size and indicate clearly to which ion they refer to.

Answer: We thank the reviewer for this suggestion. Figure 1 has been modified and updated in this revised manuscript. However, it is difficult to add more notes directly to the figure; it will be very distractive. What we have improved in this revised work is that we describe the experiment and data with detailed information.

  1. Consider changing the information contained in phrases 188-190 to the materials and methods sections.

Answer: The statement in the phrases 188-190: “Starting at the age of 8 weeks, the animals were treated with high doses of ERGO solution (25-50 mg/Kg) formulated in double-distilled (dd) water via oral gavage three times a week over the course of eight weeks”. We think it should be in its original place since it provides information relevant to the timeline of the therapy.

4.Consider changing the materials and methods section of place to right after the introduction.

Answer: We have to admit that there is no consistent order for this special issue in the journal of Pharmaceuticals. As we refer to the template of this special issue: “ A convenient route to new (Radio)fluorinated and (radio)Iodinated cyclic tyrosine analogs by Maria Noelia Chao et al. which also put the materials and Methods after the Discussion. Thus, we request to keep our order the same.

  1. In phrase 269-270 I suggest keeping “8 weeks” instead of “2 month-old”.

Answer: In this phrase, we describe the age of animals at the start of the therapy. And throughout the manuscript, we stated “2-month-old”. And thus, it will be more consistent if we keep it all in that way.

  1. In figure 3, please present the immunohistochemistry images for the control brain slices.

Answer: It is an established fact that control mouse (C57BL/6) brains do not have amyloid plaques. The objective of the data shown in Figure 3 K-S is to show the difference in amyloid levels between treated and non-treated cohorts. Further, Figure 3 has 24 subfigures; it will be very difficult to view if we squeeze more figures in.

  1.  I suggest improving image 4 by indicating which markers are labeled in the images for better readability (I believe such captions are standard and include the marker + counter-staining used; e.g. DAPI/GFAP or DAPI/IBA1).

Answer: We thank the reviewer for this idea. This new information is found in Figure 4 of this revised manuscript.

  1. Consider reducing figure 5 in size.

Answer: we reduced the size as suggested in this revised manuscript.

  1. In phrase 382, the symbol for Iron was written twice.  Please address this.

Answer: We thank the reviewer for pointing out this typographical error. It is now fixed in the revised manuscript.

  1. In your study no controls of anti-ROS and/or metal chelators were use to understand fully the mechanism of ERGO on the brain.  Can you comment on this in your manuscript?

Answer: We do not understand it means anti-ROS. But if you look at Figures 1 F and G, in the red curves, in the absence of ERGO, the non-fluorescence dye (indicator) was oxidized by free radicals and emitted a strong fluorescence signal. In the presence of increasing concentrations of ERGO, the fluorescence signal decreases in a concentration-dependent manner. We described very much in detail ROS and metal scavenging assays in the revised manuscript.

  1. Please present the variation of weight for the mice populations used in this study.

Answer: We didn’t weigh animals frequently during the therapy since there were no signs of sickness or weight loss among the animals.

  1. Please clarify whether the mice had free access to food and water.

Answer: As stated in the manuscript, our animal work is approved by Vanderbilt IACUC. All animal facility at Vanderbilt is restrictedly managed professional team, including veterinarians, and animals have free access to food and water.

  1. Please include the DNA analysis of the mice populations used (referred in section 3.5).

Answer: We described the genotyping very carefully in section 4.5. But it is very unusual to present the DNA analysis in this manuscript. To our knowledge, no one should do that since this is not a genotyping data of a new mouse model. This 5XFAD has been known for more than a decade; there are many of information out there and from Jackson Lab if you want to know. My lab has maintained this clone for over a decade.

This manuscript is a resubmission of an earlier submission. The following is a list of the peer review reports and author responses from that submission.

Round 1

Reviewer 1 Report

In the manuscript entitled “Longitudinal consumption of ergothioneine reduces oxidative stress, amyloid plaques, and improves cognition in the 5XFAD mouse model of Alzheimer's disease” the authors tested ERGO for its beneficial effects in an animal model of AD. The study lacks clinical relevance since they used young mice without clear signs of cognitive impairment, which lowers the interest significantly in the results and conclusions. The following drawbacks have been found in the manuscript:

  1. In the abstract indicate the age of the young AD mice, and what was the rationale for using young mice in this study.
  2. In the abstract, does modest improvement means statistically significant improvement or just a trend?
  3. It is anticipated that young animals will not have cognitive decline. Why was the rationale for testing cognition in young mice. An older cohort of mice of at least 5 to 6 months needs to be added to the study.
  4. The abstract states that ERGO is not present in animals. However, the introduction states that ERGO decreases with age. Please clarify.
  5. What cells were used in the 2.2 experiment? Please add to the results.
  6. Were two doses used in the study? 25 and 50 mg/kg?
  7. The behavioral tests without an aged affected group decrease the enthusiasm in this study.
  8. Add the age of the animals to section 4.5.
  9. Add to the methods the stereological procedures used to quantify the IHC data.
  10. If the idea of the study was to use ERGO as a prophylactic, then the animals should have been treated since a young age and let to survive until the control group started to show cognitive deficits.

Reviewer 2 Report

In manscript#1625626, the authors evaluated the beneficial effects of Ergothioneine (ERGO) on disease progression-related changes in behavior and neuropathology in a mouse model of Alzheimer's disease (AD). They assessed cognition performance via a set of classical behavioral tests, and evaluated the Abeta accumulation, ERGO uptake, and glucose metabolism in the brain via neuroimage techniques. Immunohistochemical staining on brain sections has been used to examine the neuropathological changes in the presence or absence of ERGO treatment. They found that ERGO treatment strikingly reduces amyloid plaques, suppresses oxidative stress, and improves glucose metabolism, together with a modest improvement in cognition deficit.

Although their study has some value on preclinical tests on potential therapeutics for this neurodegenerative disease, some issues, majorly on their experimental design, have devalued the significance of their findings in this study. Major points are listed below:
1) While 5XFAD mice from Jackson Laboratories are commonly used in the AD research community, strain information and genetic background of their AD mice should be provided. 5xFAD mice have been reported to have a sex difference in disease progression and neuropathology. Sex information of the mice in their morphological studies should be provided. In addition, a larger sample size (n>3) would be better if both sexes were included in the study.

2) When examining the anti-oxidative stress effect of ERGO, the authors used HELA cells. However, other commercially available human neural cells will be better able to mimic AD conditions in the brain, including oxidative stress.

3) Only a single dosage of ERGO has been used in all the experiments in this study with no rationale for their dosage selection. The reviewer believes that another dosage will help them examine dose-dependent effects in mice and better interpret their experimental results.

4) Some information about setting the control group is missing. For example, it is unknown what kind of chemical has been used to evaluate the metal chelation effect of ERGO. Also, a dose-dependent effect of ERGO on metal chelation should be examined.

5) To detect the effect of ERGO oxidative stress via DCFH-DA fluorescence signals, it is necessary to rule out the possibility that ERGO suppresses the activity of endogenous esterase before making conclusion that reduction DCFH-DA fluorescence signals is a result of scavenging ROS.

6) After treatment, the concentration of ERGO in the mouse brain should be examined in order to evaluate whether the concentration of ERGO in the brain of treated mice can effectively chelate metal or suppress ROS.

7) In Figure part. Although the neuroimaging data indicated an overall change in ERGO/Abeta plaque/FDG uptake in the mouse brain following treatment, the correlations between the ERGO distribution and brain pathology (i.e., astrocyte activation, Abeta accumulation, glucose metabolism) are still not known. Additional examination of the spatial profile in the brain section may help determine the relationship between brain ERGO and neuropathology.